# A Hybrid Planning Approach Based on MPC and Parametric Curves for Overtaking Maneuvers

**DOI:** 10.3390/s21020595

**Published:** 2021-01-15

**Authors:** Ray Lattarulo, Joshué Pérez Rastelli

**Affiliations:** Department of Automotive in the Industry and Transportation Division, Tecnalia Research and Innovation, 48160 Biscay, Spain; rayalejandro.lattarulo@tecnalia.com

**Keywords:** Automated Driving Systems, overtaking, path planning, speed profile, Model Predictive Control, Cooperative Connected Autonomous Vehicles

## Abstract

Automated Driving Systems (ADS) have received a considerable amount of attention in the last few decades, as part of the Intelligent Transportation Systems (ITS) field. However, this technology still lacks total automation capacities while keeping driving comfort and safety under risky scenarios, for example, overtaking, obstacle avoidance, or lane changing. Consequently, this work presents a novel method to resolve the obstacle avoidance and overtaking problems named Hybrid Planning. This solution combines the passenger’s comfort associated with the smoothness of Bézier curves and the reliable capacities of Model Predictive Control (MPC) to react against unexpected conditions, such as obstacles on the lane, overtaking and lane-change based maneuvers. A decoupled linear-model was used for the MPC formulation to ensure short computation times. The obstacles and other vehicles’ information are obtained via V2X (vehicle communications). The tests were performed in an automated Renault Twizy vehicle and they have shown good performance under complex scenarios involving static and moving obstacles at a maximum speed of 60 kph.

## 1. Introduction

Problems related to traffic have been of major concern in societies for many decades, especially those related to maximizing safety and traffic efficiency [1]. The authors of [2] have related traffic problems with conditions for risky maneuvers such as overtaking and obstacle avoidance, and both maneuvers can be considered complex and risky scenarios.

The National Highway Traffic Safety Administration (NHTSA) has calculated at least 1 person dies and more than 50 people are injured in a traffic accident every 100 million miles traveled [3], and 94% of those cases are related to human mistakes [4]. This is a problem to solve in the upcoming years because evidence has shown an increasing death rate associated with traffic accidents until 2030 [5]. Consequently, a study proved the population is concerned about the safety and security of the Automated Vehicles (AV) [6]. This is due to the promising benefits in terms of safety and passengers’ comfort [7].

One key component of this technology is the software embedded in these intelligent systems. Accordingly, AV’s decision and control processes have been divided into three task levels: (i) strategic level associated with navigation, (ii) tactical with the trajectory generation, and (iii) operational level or tracking of the trajectory planned [8]. Risky maneuvers as lane changing, overtaking, and obstacle avoidance are related to the tactical and operational level in terms of the trajectory generation and tracking [9,10,11].

There are different approaches to solve the lane change-based problems. There are two general groups to tackle dynamic environment planning, which are: (i) sampling-based methods (trajectory candidates) and (ii) Optimal Control Problem (OCP) methods [12]. A considerable number of works were executed in sampling-based approaches but general solutions using OCP have not been studied enough, especially the benefits of MPC with the currently available solvers (e.g., ACADO [13]). Other approaches are based on offline optimization but with an extra stage including dynamic obstacles [14]. A good review of the motion planning techniques for lane-change and obstacle avoidance for highway scenarios is presented in [15].

Generally, sampling-based methods are good enough for tackling simple cases of lane change-based problems (top part of Figure 1), but more general solutions can be obtained using real-time trajectory planning based on MPC (bottom part of Figure 1) considering obstacles on route, other participants and unexpected conditions as part of the problem constraints.

Some previous works from the authors have considered parametric curves and MPC, but for the controller point of view [16]. Only simulation results without moving obstacles were considered in these previous works. In this work, our approach is extended to combine the passenger’s comfort associated with the smoothness of Bezier curves and MPC to react against unexpected conditions. It is a trajectory planning method, which combines the benefits of the parametric curves of Bézier, resolving the problem of trajectory smoothing (nominal trajectory) [17], and a fast linear-MPC set in parallel, for maneuvering, in case of obstacle avoidance or overtaking. The MPC has used a decoupled dynamics model based on integrator chains because they can be resolved in short computation times in case of unexpected conditions [18]. In this work, real tests, using communication protocols, have been implemented for the overtaking maneuver with two connected and automated vehicles in complex scenarios with static and moving obstacles (at 60 kph).

The rest of the paper is organized as follows: Section 2 contains a brief summary of previous work in this area. Section 3 explains the hybrid trajectory planning method combining Bézier curves and MPC. Section 4 has all the information related to the control architecture used, the tracking controllers used, the MPC solver and the test vehicle along with the proving ground. The experimental results are presented in Section 5. Section 6 describes the conclusions and discussions of our approach.

## 2. Review on Vehicle Motion under Unexpected Conditions

The overtaking and obstacle avoidance problems have been strongly studied in ADS’ literature [19]. These maneuvers are a key point of AV because: (i) they can improve traffic flow, (ii) reduce the impact of slow speed vehicles, and (iii) improve the comfort of the passengers. However, there are risks during the execution, demanding a good driver capacity to anticipate reactions of the other vehicles involved.

Some authors have studied overtaking maneuvers as a three-phase movement: (i) departing from the original lane, (ii) cruising in the opposite lane to overtake the other vehicle, and (iii) returning to the original lane [20,21,22]. In this three-phase approach, space and time gaps are temporally blocked for the maneuver execution with other moving vehicles; these gaps are proven to be of difficult calculation for vehicle automation [23,24] and the other vehicles could not understand the blockage of the space for cooperative maneuvers.

This knowledge of separation in stages has been brought to the ADS field since the early days, for example, works done in automated overtaking using fuzzy logic principles [25], path generation for lane change-based maneuvers using parametric curves [26], among others. Nevertheless, they are mostly used for solving simple cases under ideal conditions of overtaking in straight segments with a maximum number of three vehicles.

Newer tendencies in ADS try to use the benefits of Machine Learning (ML) architectures in AV to simplify the process of imitating human driving-style. For example, ML approaches have been successfully deployed in the area of obstacle recognition and classification under real-time driving conditions.

Another tendency is the use of real-time optimization methods because they have a good capacity to solve the problem without programming specific cases or using an extensive dataset [14]. Another approach uses trajectory planning algorithms that are able to provide safe, human-like and comfortable trajectories by quintic Bézier curves [27]. Other studies have included a Potential Field (PF) around surrounding obstacles, generating an optimal trajectory to drive the vehicle stably [28] considering curved roads as well [29].

Dynamic programming, indirect and direct methods are the three general classes mostly used to solve optimization problems. The first one is strong in non-convex problems but it demands a big amount of computational time, making real-time computation extremely difficult. The second one shows efficient results but with a trade-off of high difficulty in introducing constraints. In general, direct methods, for example, Sequential Quadratic Problems (SQP), are strong enough against non-linear problems and it is relatively easy to introduce constraints in them [30]. Most of the MPC solvers use the SQP method [31], and some applications have been made with Artificial PF plus MPC to resolve the planning problem [32], sharing control between driver and vehicle under overtaking maneuvers [33], optimization during the merging process based on acceleration and jerk [34], distributed MPC solving highways overtaking and lane change problems [35,36]. Most of them, in simulation environments [37], or real platforms at low speeds less than 15 kph [38].

A recent study has proved Bézier curves are a reliable solution for an optimal trajectory generation [27,39]. The author of that approach has proposed a spatio-temporal safe corridor, which is a step previous to the trajectory generation. The authors have described a collision-free corridor where trajectory optimization is generated into it. Nevertheless, this corridor has segmented the description of the obstacles in static and dynamic ones that lack generality under real circumstances.

The topic of real-time risk-assessment is of interest to design safe AV. The approach presented by [40] has established a system for evaluating the risk of a maneuver based on the information produced by a previous trajectory and a collision evaluation. This method is used in the case of detecting a possible risk, which overpasses a threshold, starts an emergency maneuver based on motion candidates, which are evaluated using the same risk-assessment approach. A nominal planner is running in parallel in the case of an obstacle-free road.

Analyzing the state-of-the-art, there are some gaps in the technology in terms of the generation of a real-time trajectory that can ensure safety while managing unexpected conditions without hard-coding a bunch of specific cases. A good solution is to propose a smooth trajectory generation method (Bézier curves) with the capacity of generating a real-time response in the case of unexpected conditions (MPC), which is safe by a binary-evaluation of the risk over the maneuver (collision checking).

## 3. Hybrid Planning Approach

The approach presents a method to resolve the overtaking and obstacle avoidance maneuver. This hybrid planning method combines a nominal trajectory generated with Bézier curves (parametric curves) and the maneuver optimization provided by an MPC. No specific conditions have been considered, and the problem is completely solved using MPC constraints and reference manipulation.

A general vision of the approach is presented in Figure 2. Vehicle (1) represents the ego-vehicle, (2) and (3) are possible obstacles or other vehicles blocking the nominal behavior of the ego-vehicle. The dashed line represents the nominal trajectory blocked by vehicle (2) and in the pointed line is the hybrid trajectory (a combination of the Bézier nominal trajectory and the MPC). The generated trajectory is validated with a collision verification (Section 3.4) using the information of the previous iteration.

This nominal trajectory was inspired in a previous work [41]. All the conditions for the Bézier-based planner are explained and the maneuver planner using MPC is explained in Section 3.1 and Section 3.2, respectively. Section 3.3 has the information related to the merging process of the nominal and MPC trajectories.

### 3.1. Bézier Nominal Trajectory

Bézier curves are a type of parametric curves widely used in ADS in the last years [26,42,43], due to their useful properties for trajectory planning. The general expression used to define this set of curves is:(1)B(t|n,P0,…,Pn)=∑i=0nbiPi,bi=niti(1−t)n−i
where $P_{i}\in R^{2}$ are control points, n∈N is the curve order, t∈[0,1] is the curve construction parameter and $b_{i}$ is known as Berstein Polynomial. In general the formula above is defined in this case as a function $\left\{B:R\to R^{2}\right\}$ due to its construction in the <x,y> plane.

Figure 3 depicts the control points used for curve construction, the total amount of these points will be n+1. Relevant properties are:The generated curve starts at control point $P_{0}$ and finishes in $P_{n}$.The direction in the start and end of the trajectory are defined by $\vec{P_{0}P_{1}}=v_{0}$ and $\vec{P_{n-1}P_{n}}=v_{n-1}$ (Figure 3).The curve will lay into a convex hull formed by the n+1 control points.They are $C^{n}$ (geometrical) and $G^{n}$ (curvature) continuous.The amount of changes in curvature concavity is proportional to the changes of the vectors in $P_{n}$.

These curves were used for the nominal trajectory planning as it is explained in [41]. This previous approach was based on concatenating Bézier curves, which are generated in the case of intersections, and roundabouts (entrances and exits). These segments are combined with line segments and in the case of a roundabout entrance with the exit with an arc segment proportional to the roundabout curvature radius preserving $C^{2}$ and $G^{2}$ continuity.

The generated trajectory is fully contained in the path lane, keeping safety and feasible solutions for the vehicle. Additionally, the total distance of the generated trajectory is fixed considering a vision horizon. Nevertheless, in these experiments the nominal trajectory was pre-computed offline, in a newer version of this method, the nominal trajectory is computed online [14].

### 3.2. MPC Maneuver Planning

In ADS, the use of MPC for decision and control have been mainly based on the kinematic bicycle model and the dynamic model of the vehicle, which includes forces over the tires [44]. These models have non-linear components that improve the model approximation, but they demand more computation time (non-linear MPC). On the other hand, better computation performance can be reached using linear models, for example, point mass (integrator chains), with the trade-off of compromising the approximation in case of tight curves (this is tackled by the nominal planner based on Bézier).

A linear model of the vehicle was selected to find an optimal solution, with a fast response time. The model used is a decoupled integrator chain model for longitudinal and lateral dynamics, which is mostly related to holonomic systems; nevertheless, the constraints were manipulated to provide non-holonomic properties [45]. Additionally, the type of maneuvers executed are lane-change maneuvers, i.e., obstacle avoidance and overtaking, which are not compromised by the model precision.

#### 3.2.1. Longitudinal Model

The proposed model was a triple integrator chain based on jerk, as it is shown in:(2)$$d_{lon}=\;\int \int \int \;j_{lon}\left(t\right)\;dt^{3}$$
where $d_{lon}$ is the longitudinal distance, $j_{lon}$ is the longitudinal jerk. Additionally, from this integration can be obtained longitudinal speed $v_{lon}$ and the longitudinal acceleration $a_{lon}$. This model can be written using the linear state-space representation:(3)$$\begin{array}{c} {\dot{X}}_{lon}=\left[\begin{array}{c} {\dot{d}}_{lon}\\ {\dot{v}}_{lon}\\ {\dot{a}}_{lon} \end{array}\right]=\left[\begin{array}{ccc} 0 & 1 & 0\\ 0 & 0 & 1\\ 0 & 0 & 0 \end{array}\right]\left[\begin{array}{c} d_{lon}\\ v_{lon}\\ a_{lon} \end{array}\right]+\left[\begin{array}{c} 0\\ 0\\ 1 \end{array}\right]j_{lon} \end{array}$$

The output of the MPC longitudinal portion is a smooth speed profile. Nevertheless, it depends on the acceleration produced by the throttle and brake. Moreover, these acceleration changes must be smooth to avoid unexpected movement while the vehicle is cruising. The smoothness of its rate of change is the reason to include acceleration and jerk as part of the problem formulation.

All the states and control variables are bounded constraints, which is a special case of inequality constraints. Those constraints are:(4)$$\begin{array}{cc} 0\leq d_{lon} & \leq D_{veh_{front}}\\ 0\leq v_{lon} & \leq v_{lon_{max}}\\ a_{min}\leq a_{lon} & \leq a_{max}\\ -|j_{max}|\leq j_{lon} & \leq |j_{max}| \end{array}$$
the jerk has been included in the model to add the component of comfort and this value is limited to a maximum value of jerk $j_{max}$ (further information of this parameter and its relationship with comfort is presented in [46]). The acceleration is bounded between the limits of maximum deceleration $a_{min}$ and a maximum acceleration of $a_{max}$. The speed is bounded between 0 and the maximum permissible speed $v_{lon_{max}}$. The longitudinal displacement is bounded to avoid a rear-end collision with a vehicle located at a distance $D_{veh_{front}}$. The set of longitudinal boundary constraints will be named $h_{lon}\left(t\right)$ in Section 3.2.3.

The maximum permissible speed is calculated using the formula of total acceleration and an approximation for small variations in longitudinal ($a_{x}\approx 0$) and vertical ($a_{z}\approx 0$) accelerations, as well as the relation between the centripetal acceleration and longitudinal speed:(5)$$\begin{array}{cc} a_{\omega } & =\sqrt{{\left(1.4a_{x}\right)}^{2}+{\left(1.4a_{y}\right)}^{2}+{\left(k_{z}a_{z}\right)}^{2}}\approx 1.4a_{y}=1.4\left(v_{lon}^{2}\kappa \right)\\ v_{lon} & =\sqrt{\frac{a_{\omega }}{1.4\kappa }} \end{array}$$

$a_{\omega }$ is the total comfortable acceleration felt by a human and it has some limits described in [47]. The $a_{x}$, $a_{y}$ and $a_{z}$ are, respectively, the longitudinal, lateral, and vertical accelerations of the vehicle. The curvature of the path is denoted with κ. Introducing the variable $v_{limit}$, which is the speed limit of the path, then the permissible speed is considered as the minimum between the road limit and the comfort condition:(6)$$v_{lon_{max}}=\min\left\{v_{limit},\sqrt{\frac{a_{\omega }}{1.4\kappa }}\right\}$$

#### 3.2.2. Lateral Model

The model is a double integrator chain, which is described by:(7)$$d_{lat}=\;\int \int \;a_{lat}\left(t\right)\;dt^{2}$$
$d_{lat}$ is the lateral offset (distance) with respect to the nominal trajectory (Bézier) and $a_{lat}$ is the lateral acceleration. Additionally, $v_{lat}$ is obtained during the integration process and it is the lateral speed. This model can be represented with a linear state-space representation:(8)$$\begin{array}{c} {\dot{X}}_{lat}=\left[\begin{array}{c} {\dot{d}}_{lat}\\ {\dot{v}}_{lat} \end{array}\right]=\left[\begin{array}{cc} 0 & 1\\ 0 & 0 \end{array}\right]\left[\begin{array}{c} d_{lat}\\ v_{lat} \end{array}\right]+\left[\begin{array}{c} 0\\ 1 \end{array}\right]a_{lat} \end{array}$$

The lateral output of the MPC will be a smooth lateral offset, which is adapted according to the presence of obstacles on the road. Nevertheless, this value depends on the current vehicle speed. The lateral speed permits to constrain this offset, and its relationship will be studied in future works. The lateral acceleration was added to avoid unexpected speed changes.

All the states and control variables are bounded with the constraint set:(9)$$\begin{array}{cc} -\frac{1}{2}Rw+\frac{1}{2}W\leq d_{lat} & \leq \frac{3}{2}Rw-\frac{1}{2}W\\ -|v_{lat_{max}}|\leq v_{lat} & \leq |v_{lat_{max}}|\\ -|a_{lat_{max}}|\leq a_{lat} & \leq |a_{lat_{max}}| \end{array}$$
Rw is the road width, *W* is the vehicle width. The approach assumes Rw>W. The lateral acceleration and speed are bounded with constant values and the lateral offset varies from the bounds of the opposite lane and the current lane. The variable $d_{lat}$ is dynamically defined during execution, but its limit values are shown in the inequality, considering the road’s width Rw and the vehicle’s width *W*. The lower bound is defined by $-\frac{1}{2}Rw+\frac{1}{2}W$, and the upper bound $\frac{3}{2}Rw-\frac{1}{2}W$. For the approach, we have considered the presence of the second lane of width Rw, on the left side; this condition generates the component $\frac{3}{2}Rw$. The movement of the lateral bounds is explained in Section 3.4. The lateral bounded constraints are named as $h_{lat}\left(t\right)$ in Section 3.2.3.

#### 3.2.3. Optimization Function

The optimization process is resolved using a Quadratic Problem (QP) formulation, where it minimizes the function Φ(X(t),u(t)), which is a quadratic cost function of the state vector $X\left(t\right)={\left[X_{lon}\;X_{lat}\right]}^{T}$ and the control vector $u={\left[j_{lon}\;a_{lat}\right]}^{T}$. This problem can be summarized as:(10)$$\begin{array}{cc} minimize\; & \Phi (X(t),u(t))\\ subject\;to\; & h_{lat}\left(t\right),\;h_{lon}\left(t\right) \end{array}$$
where $h_{lat}\left(t\right)$ and $h_{lon}\left(t\right)$ are the lateral and longitudinal bounded constraints, respectively. The optimization function is defined by:(11)$$\begin{array}{cc} \Phi (X(t),u(t)) & ={\left(d_{lat}\left(t\right)-d_{ref_{lat}}\left(t\right)\right)}^{2}\\ \; & +{\left(v_{lon}\left(t\right)-v_{ref_{lon}}\left(t\right)\right)}^{2} \end{array}$$

This problem will have linear bounded constraints due to the performances of other participants (e.g., blocking the route or driving at slower cruising speed). In this sense, the convex optimization function and the inequality constraints result in a convex problem with a feasible and fast solution. In the case of varying the lateral (offset) and the longitudinal (distance and speed) constraints, the linear inequality constraints will be transformed into non-linear constraints, which could diverge in unfeasible solutions. Mechanisms for detection and mitigation of the unfeasible solutions have been implemented (see Section 3.4).

### 3.3. Combination of Both Trajectories

Both trajectories are combined considering the method described in [16], which is presented in Figure 4. The diagram uses as input the information of the vehicle (ego vehicle), the Nominal Trajectory Generator (based on Bézier curves, Section 3.1), and other vehicles and obstacle information received through the perception and/or the communication modules. These inputs are sent through a buffer to the decision modules, as in [42]. In this work, the decision block for the overtaking maneuvers is divided into three modules: MPC integration chains (explained in Section 3.2), MPC Calculator for the lateral and longitudinal references (explained in Section 3.4, and the *Nominal Trajectory Calculator*.

The *Nominal Trajectory Calculator* module uses this information from perception and communication modules to generate the control variables: lateral error, angular error, curvature, and the speed setpoint. The lateral error is the deviation of the vehicle front and the nominal trajectory. The angular error is the angle between the vehicle’s orientation and the nominal trajectory. The curvature is the inverse of the turning radius in each segment of the trajectory, as it is defined in [17]. The MPC integrator chains add a new offset for the *Nominal Trajectory Calculator*. This modified trajectory is sent to the control blocks (both lateral and longitudinal controllers, as is shown in the lower right part of Figure 4).

The same inputs are used as well in the *MPC calculator* module, which is in charge of modeling the constraints and the reference of the MPC. This module uses the lateral and longitudinal references, the lower and upper bounds (see Equation (Equation 9)), and the information from the obstacle and other vehicles around, to give the new reference and constraints to the MPC. Additionally, the MPC output (control vector explained in Section 3.2.3) is integrated to generate the future states (i.e., position for the lateral part of the trajectory and speed and acceleration for the longitudinal part of the trajectory), which are used for the collision verification.

The system outputs consider the minimum speed setpoint between the one obtained from the MPC and the one provided by the nominal trajectory. This is a safety consideration avoiding rear-end collisions. In the case of the lateral domain, the lateral displacement of the first MPC state is added to the nominal lateral error, generating a possible lane-change maneuver.

This combination of both approaches permits to lead the vehicle safely through intersections, roundabouts, and merging using the nominal trajectory; the MPC will be in charge of the dynamic speed adaption (in all situations), overtaking, and obstacle avoidance maneuvers.

### 3.4. MPC Calculator: Constraints and Lateral/Longitudinal References

Lane-change based maneuvers depend on situations that constantly change (e.g., moving totally to another lane or braking in emergency situations) and constraint satisfaction (safety) is a demand on vehicles (safety-critical systems). In these terms, the MPC calculator will be in charge of manipulating the MPC reference and constraints according to the nominal trajectory and obstacles on the road.

This module collects the information generated in the MPC previous iteration, and future vehicle states are reconstructed with it (Figure 4). Propagation of the vehicle over the nominal lane and the opposite lane is done as presented in Figure 5a. The nominal trajectory is used as a basis for the propagation using the integration of the longitudinal jerk in time several samples. The propagation over the opposite lane is generated with the one executed over the nominal trajectory and the projection over the center of the opposite lane at a distance of Rw. In the case of more than two opposite lanes (right and left side of the nominal lane), the future vehicle states are propagated over all of them using the nominal trajectory as a basis.

In this approach, the obstacles are transmitted via communications (V2X) using a piece of third party equipment and connecting it to the control interfaces via a TCP socket [48]. Nevertheless, the data transmitted can be adapted to obstacles obtained via perception due to the variables used. The longitudinal speed and acceleration, current position, orientation, length, and width are needed to model the obstacles.

A collision evaluation is executed over each future vehicle state (propagated ones) and the propagation of other vehicle positions using a kinematic model. An evaluation of vehicles’ segments intersection is executed and in case of at least one intersection, that space and time will be tagged as collision (Figure 5a).

In the case of a collision in the opposite lane, the lateral bounds move to the nominal-lane. In the case of a collision in the nominal-lane, the lateral bounds move to the opposite lane. The reference of the lane offset is considered the first bound change. An example is shown in Figure 5b, where the vehicle drives in the opposite lane; however, a change to the nominal-lane was done due to a possible collision detected in that lane from time 1.5 and 2.5 s. This reference is moved to the nominal lane (red line in the top part of Figure 5b).

In the case of detecting a blockage in the lanes, the vehicle will limit the longitudinal distance constraint up to the blockage distance. Another example is shown in Figure 5c, where the ego vehicle has limited the distance to 10 m, after 2.5 s, because a possible collision could take place. Moreover, the example presents two vehicles blocking the lane. In this case, the vehicle keeps the reference of the MPC in the lane.

In the case of verifying unfeasible conditions (states out of the bounds), the ego vehicle moves the reference speed to 0 from the moment of the detection of an unfeasible condition. Figure 5d shows an unfeasible condition after 2.5 s, which generates a speed reduction to keep a safe inter-vehicle distance for the automated vehicle.

## 4. Experiment Set-Up

This section explains the experimental set-up, which includes: the tracking controllers used during the tests, the MPC solver, the prototype AV, and the test tracks. Moreover, the control architecture was the one presented in [42].

### 4.1. Tracking Controller Used

A control law, based on lateral and angular error and the curvature of the trajectory, was selected for the lateral control (vehicle steering) [43,49]. This controller has the problem of the trajectory point used for calculating the control variables. Figure 6 depicts this problem where it is used as a projection of vehicle lookahead distance. In the case of a short distance (top part of Figure 6), the stability of the system could be compromised at high speeds. A longer lookahead distance is used to compensate for controllers’ delay (between the control PC and the vehicle actuators), which is around 500 milliseconds, keeping the vehicle stable at high speeds (middle part of Figure 6). The real projection over the trajectory ($D_{b}$) will be shorter in bend segments.

In this sense, the method used to calculate the control variables is presented in the bottom part of Figure 6. The trajectory control point has moved a distance $D_{b}$ over the trajectory and the control variables are calculated considering this point. This mitigates the effects of control delays and, in the case of bend segments, the system has a better performance.

The distance $D_{b}$ is selected based on the delay of the actuators and control commands. Then, the distance $D_{b}$ is selected using this delay ($t_{d}$, this approach considered an actuator delay of 0.5 s) and the current speed, as:(12)$$D_{b}=v_{veh}t_{d}$$
where $v_{veh}$ is vehicle speed.

Considering the value of $D_{b}$ and the representation of Figure 6, the control variables are calculated, which are lateral error, angular error, and the curvature; they are applied to the control law defined in [43]:(13)$$Cv_{lat}=K_{lat}e_{lat}+K_{ang}e_{ang}+K_{curv}k$$
where $Cv_{lat}$ is the steering wheel command, which is in the range [−1.0,1.0]. $K_{lat},K_{ang},K_{curv}$ are control tuning gains, $e_{ang}$ is angular error, *k* curvature and lateral error is defined as $e_{lat}$. This last variable is relevant to the current approach due to the combination of nominal lateral error with the MPC lateral offset, which permits the execution of the lane-change based maneuvers.

Longitudinal fuzzy logic controllers can be easily and intuitively designed using human experience, as in [50]. A fuzzy controller was used to control the longitudinal domain. The membership functions were defined with current speed, where three different membership functions were defined for low, medium, and high, and the speed error, where the membership functions are negative, central, and positive, as in [50]. The output of the fuzzy controller is the normalized action on the throttle and brake pedals, defined in the range [−1, 0) for the brake, and in the range (0, 1] for the throttle.

### 4.2. MPC Solver

The ACADO Toolkit was the solver used for the MPC problem formulation. It is a software toolbox (library) based on C++ and it is used to solve OCP applications. It is a complete tool-chain that offers a great variety of possible configurations with efficient execution times. Some of its major characteristics are: (i) it is open-source, (ii) no external packages are needed to use this library, (iii) it is compatible with platforms such as Matlab/Simulink, and (iv) the learning process of this software is relatively fast and it has a great numbert of examples [13].

This toolkit was used to solve three typical control problems, which are: (i) offline multi-objective control problems (optimal open-loop control law), (ii) it can be used for parameters and states estimations (model identification problem), and (iii) online OCP, where an optimal control output (subject to a constraint set in a prediction horizon) is given. The configuration of the solver was multiple shooting discretization, Runge–Kutta integration of discrete states, and the SQP method to solve the minimization problem [51].

Finally, this toolbox can be used with a Matlab/Simulink interface where the AD architecture was developed [52].

### 4.3. Proving Ground and Testing Platform

Some simulation results are presented in this work using the Dynacar simulator (Figure 7 top). It uses a highly precise multibody vehicle model that has a well-designed 3D interface for vehicle monitoring. For further information related to the simulation environment, see [53].

A closed test track was used for algorithm validation under a real platform. The dimensions are 20 m wide by 85 m long. It has two lanes, roundabouts, straight segments, and additionally bend segments overlapped in the middle. The scenario can manage speeds of around 30 kph without compromising safety during test execution.

The lower part of Figure 7 shows the real vehicle platform (on the test track previously mentioned): a Renault Twizy. This is instrumented using a servo motor in the middle of the steering wheel bar, the throttle is controlled via the central ECU (Electronic Control Unit) and the brake is instrumented using a servo motor. Additionally, a Programmable Logic Controller (PLC) is installed as a gateway between the actuators and the high-level automation PC (control and decision modules in Figure 4). Some properties of this vehicle are shown in Table 1.

## 5. Experimental Results

This section presents the results obtained with the real vehicle at low-medium and at medium-high speeds in Section 5.1 and Section 5.2, respectively. Additionally, a simulation test case is presented in Section 5.3, which considers the trajectory planning under a risky scenario.

All the tests were performed considering 10 prediction samples of the MPC at a sample time of 0.5 s.

### 5.1. Low-Medium Speed Obstacle Avoidance

The results of the overtaking at low-medium are presented in Figure 8. The first part of the test track is presented in Figure 8a. The real vehicle executes tracking of the nominal trajectory (dashed line), but after detecting and evaluating the conditions over obstacle 1, the MPC starts a maneuver of lane-change avoiding a possible collision or braking the vehicle. Figure 8b depicts the capacity of the MPC of partially returning to the nominal lane, avoiding a collision with obstacle 4, and verifying the next lane-change due to obstacle 2. Then, Figure 8c shows the capacity of keeping the lane-change in bend segments with the presence of obstacles 2 and 3. The experiment finishes with the vehicle returning to the nominal lane Figure 8d.

Figure 9a depicts the lateral offset used for the maneuver planning. Between 20 and 25 s, the system is capable of generating lane changes partially completed to optimize the travel distance. This case is similar between 30 to 38 s, when the vehicle keeps the opposite lane to avoid two consecutive obstacles. This approach shows interesting results, due to the continuous optimization of the vehicle to partially execute the lane change to avoid consecutive obstacles, improving the travel distance.

Figure 9b presents the total lateral error with respect to the hybrid trajectory. This value is continuous due to the sum of two continuous trajectories (nominal and MPC). The maximum value during this experiment was 65 centimeters but normally it was under 40 per centimeters. The GPS has introduced an error of around 10 centimeters in this test due to the signal quality.

The vehicle speed is shown in Figure 9c. Some speed reduction under the limit to overcome possible unfeasible solutions are obtained during the calculation due to non-linear inequality constraints.

### 5.2. Medium-High Speed Obstacle Avoidance

The real vehicle has been tested on a different test track to increase the speed until 60 kph, which is a straight path of around 500 m long. In this case, the approach was tested in a scenario of moving vehicles in the same direction. The approach is capable of modeling the constraints around the free-collision space, which keeps safety during the execution of the overtaking maneuver. Figure 10 depicts this scenario. Figure 10d presents the maximum speed achieved during the experiment.

### 5.3. Overtaking Scenario Using Virtual Environments

A simulated test case was evaluated and the results are presented in Figure 11. The scenario was designed with two lanes in opposite directions with an automated vehicle executing an overtaking maneuver, as presented in Figure 11a. In the first part of this scenario, the automated vehicle determines the possibility to start an overtaking maneuver but later on, in the scenario, a possible collision is determined in the opposite lane, which pushes the vehicle to the nominal lane. After the dangerous condition finishes, the AV has finished the overtaking maneuver.

The returning process is demonstrated with the lateral offset in Figure 11b. The figure depicts a returning process to the nominal lane between 19 and 22 s, avoiding the risky situation and returning to the maneuver after this period. Additionally, Figure 11c shows a reduction in the maximum speed of 40 kph, avoiding a possible rear-end collision.

The algorithm has obtained reliable results under difficult situations like the one presented in this simulation test case. This type of scenario can be improved, in terms of safety, with a cooperation agreement among the vehicles. Nevertheless, more extensive tests under different situations will be performed in the future, along with multiple real platforms, to validate the statement.

## 6. Discussion and Conclusions

This work has presented a novel method to resolve overtaking and obstacle avoidance maneuvers. It considers a smooth and continuous Bézier trajectory combined with an MPC formulation. This hybrid trajectory is obtained in less than 10 milliseconds (fanless industrial PC with a Core i7-6700 processor and 16 GB of RAM) while decoupling vehicle kinematics (longitudinal and lateral). The obstacles collision evaluation demands more than 100 milliseconds, depending on the number of obstacles. Our approach has presented good results under real scenarios, considering detection and/or communication of other participants’ information. Comfort and safety (collision avoidance) have been considered in this approach.

Most of the tests have been performed with a real automated vehicle, validating the approach in a closed test track with virtual obstacles. For the bend segments, the automated vehicle has reached 30 kph in lane-change maneuver, but in longer paths, a maximum speed of around 60 kph has been reached, using communications with the other vehicles around.

Obstacle avoidance and overtaking maneuvers are supported in curve segments considering static obstacles and moving ones. The approach reduces the ego-vehicle speed to 0 kph if a blockage of the path exists.

In Table 2, a comparison of our approach with other methods is presented. MPC methods using precise vehicle models (non-linear) are known to require a high computation time, demanding a specific piece of hardware. However, this approach resolved the trajectory task in the same hardware. In the case of using Bézier curves for doing the overtaking or obstacle avoidance, the computation time can be easily reduced, but the complexity of environment evaluation increases. The MPC approach can resolve complex dynamics environments easily with the trade-off of computation time. In general, constraints for high degree polynomials (5th order) can be very complex in the formulation, but in the proposed method the constraints of the road are introduced easily. A combination of Bézier curves to generate nominal $C^{n}$ and $G^{n}$ trajectories at a low-frequency rate, with a linear MPC approach (short computation time) is considered as a good solution for automated trajectory planning. This the main contribution of this work.

Additionally, the simulation test case was included to introduce some risky maneuvers that will be tested in future works. These planned tests will include a cooperation agreement method to avoid undesired conditions, i.e., returning to the lane in the middle of the maneuver due to perturbances during the prediction computation.

## Figures and Tables

**Figure 1 sensors-21-00595-f001:**
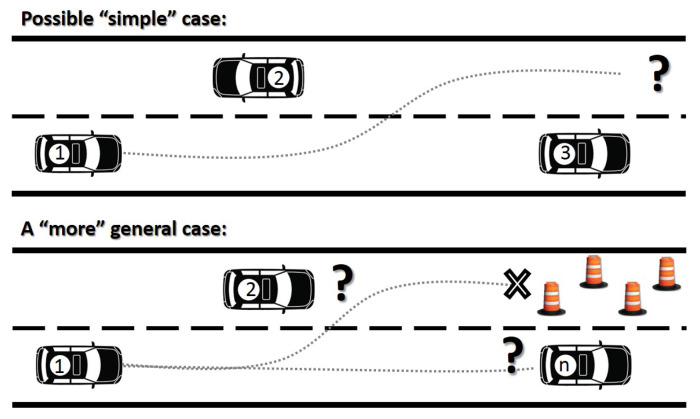
Trajectory planning problem under traffic conditions.

**Figure 2 sensors-21-00595-f002:**
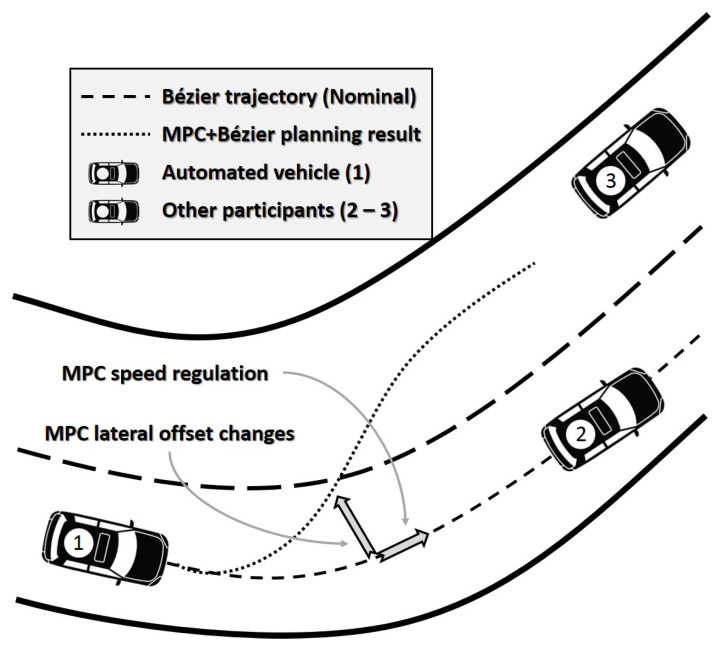
Problem segmentation in the overtaking approach.

**Figure 3 sensors-21-00595-f003:**
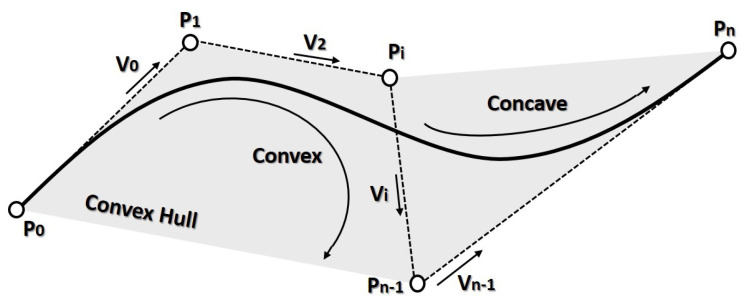
Bézier main characteristics.

**Figure 4 sensors-21-00595-f004:**
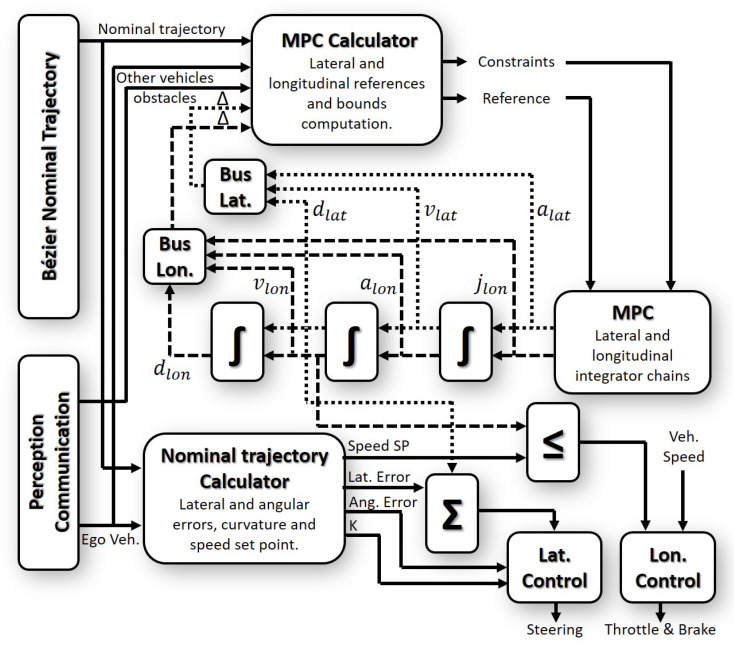
Model Predictive Control (MPC) approach block diagram.

**Figure 5 sensors-21-00595-f005:**
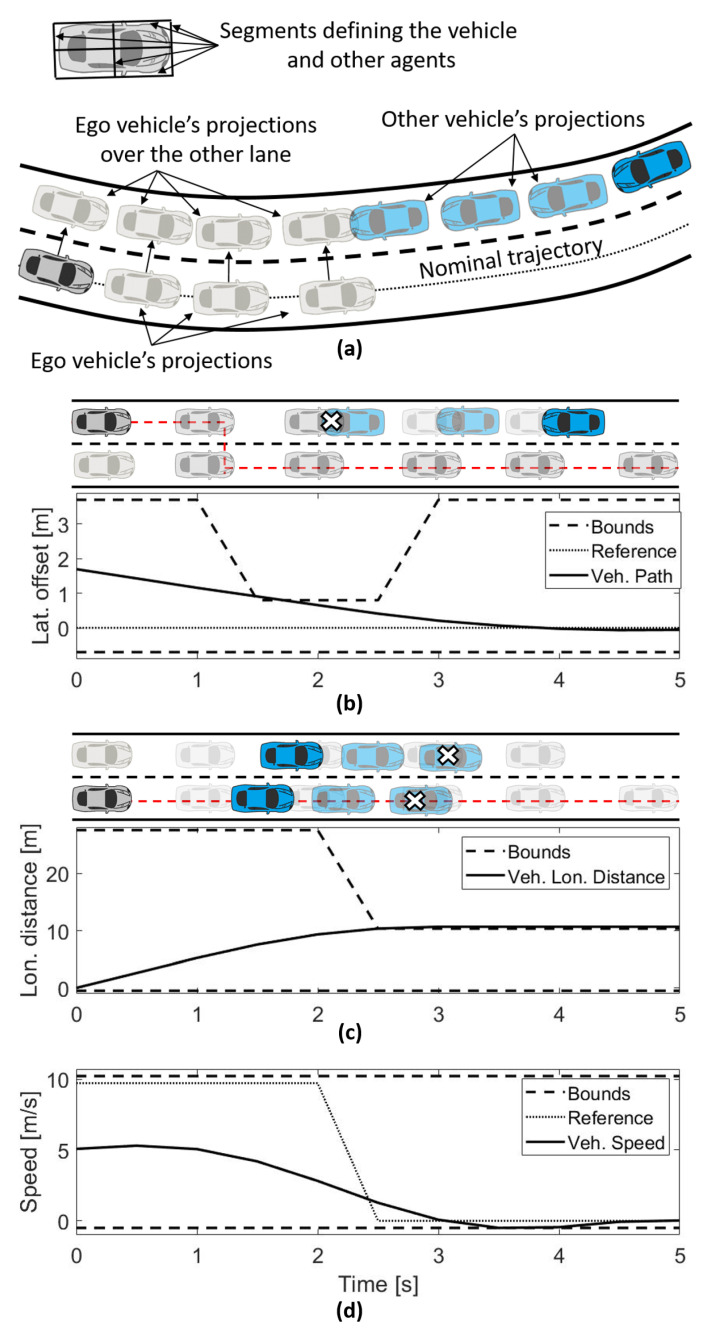
(**a**) Vehicle projections and segmentation. (**b**) Lateral offset reference and constraint manipulation. (**c**) Longitudinal position constraint manipulation. (**d**) Longitudinal speed reference and constraint manipulation.

**Figure 6 sensors-21-00595-f006:**
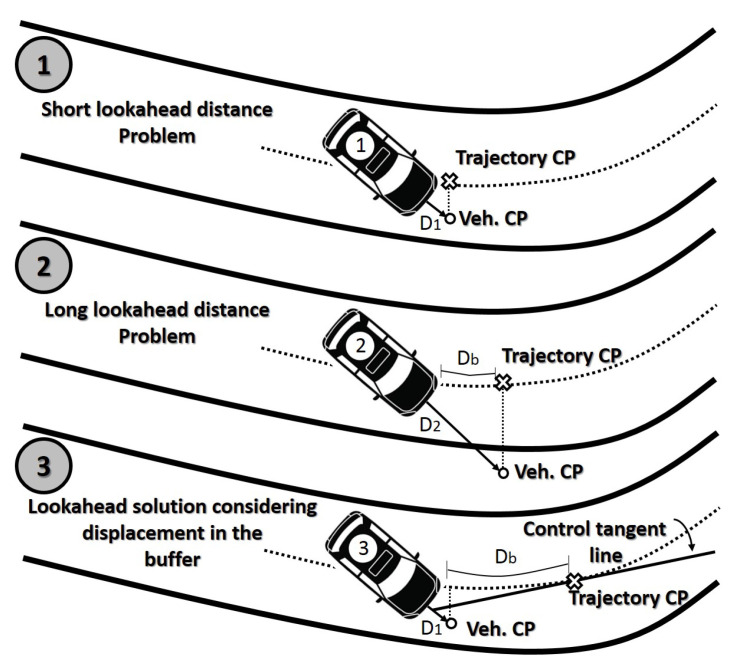
Lookahead distance: state of the problem.

**Figure 7 sensors-21-00595-f007:**
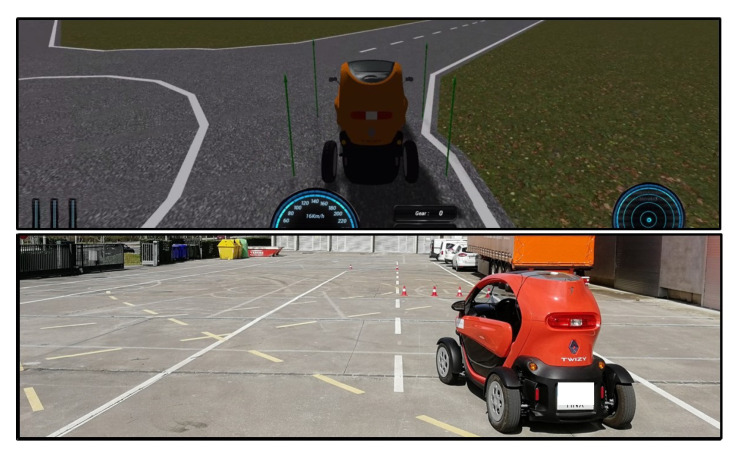
Experimental set-up: simulator and real platform.

**Figure 8 sensors-21-00595-f008:**
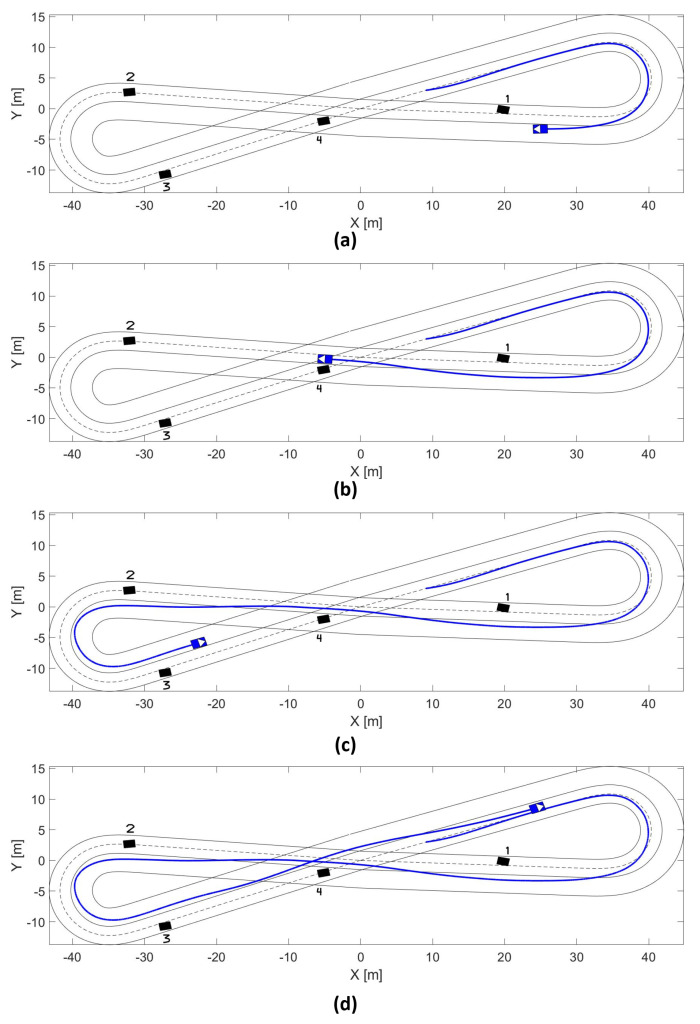
(**a**) Low-medium speed test track until 17 s, (**b**) low-medium speed test track until 22.5 s, (**c**) low-medium speed test track until 33.5 s, (**d**) low-medium speed test track until 41 s.

**Figure 9 sensors-21-00595-f009:**
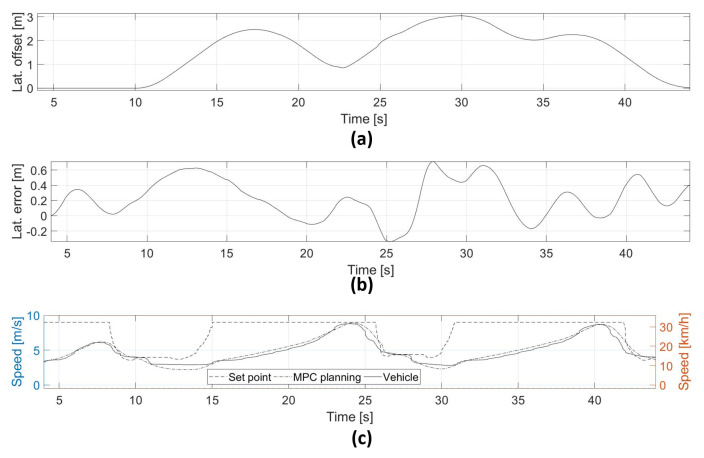
(**a**) Lateral offset used for the obstacle avoidance maneuvers, (**b**) lateral error with respect to the generated trajectory, (**c**) vehicle speed depending the test track conditions.

**Figure 10 sensors-21-00595-f010:**
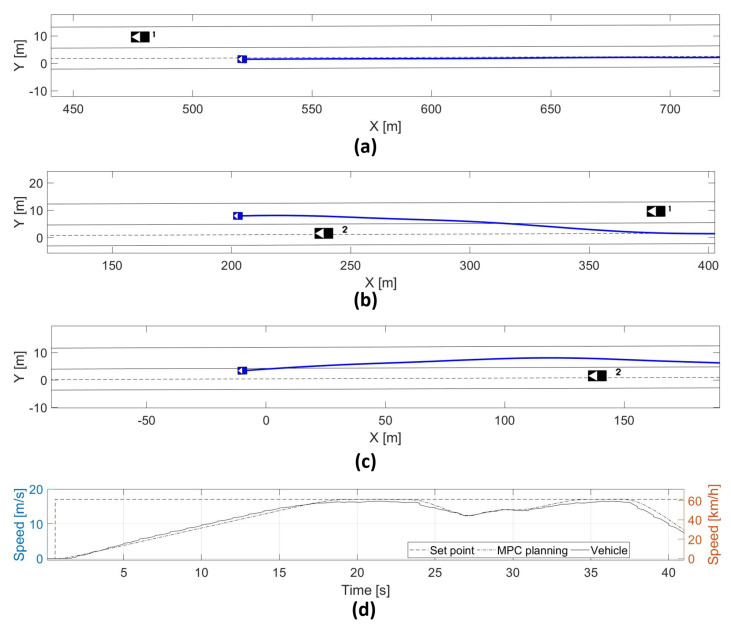
(**a**) High speed test track until 22 s, (**b**) high speed test track until 30 s, (**c**) high speed test track until 34 s, (**d**) vehicle test track in the overtaking scenario.

**Figure 11 sensors-21-00595-f011:**
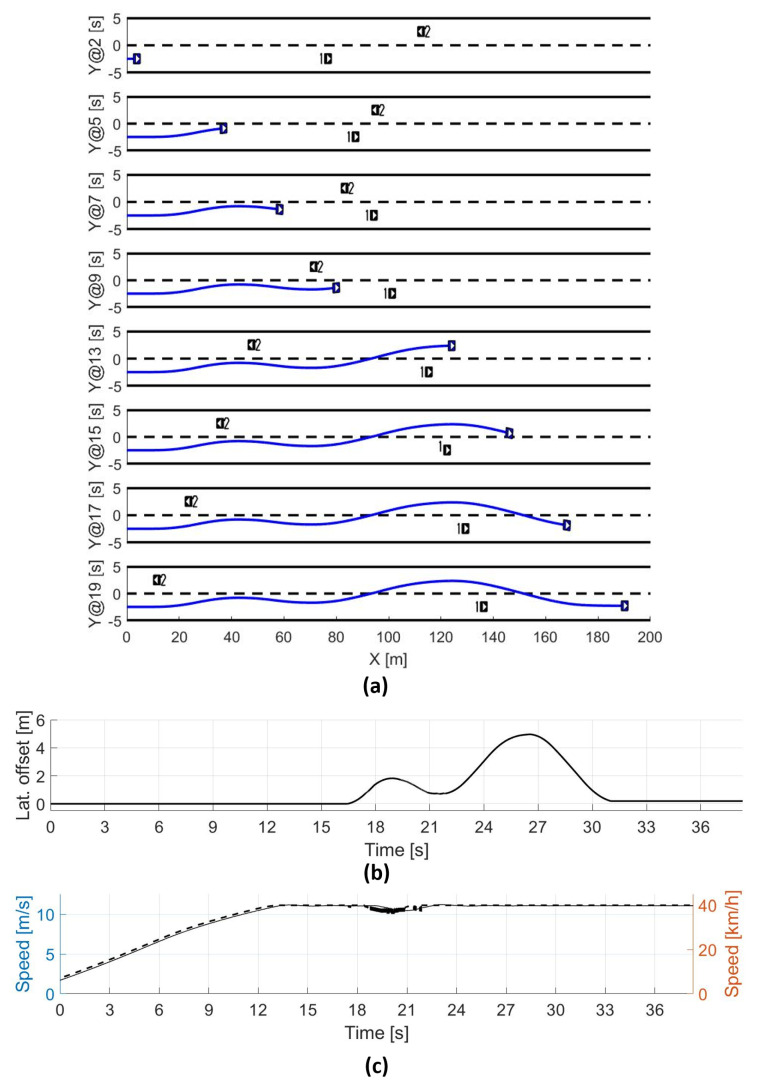
(**a**) Evolution of the scenario in time, (**b**) lateral offset for the overtaking maneuver, (**c**) ego vehicle speed during the test.

**Table 1 sensors-21-00595-t001:** Twizy general characteristics.

Property	Value
Length	2.40 (m)
Width	1.30 (m)
Max. Speed	22.22 (m/s)
Max. Acceleration	1.00 (m/s${}^{2}$)
Max. Deceleration	3.15 (m/s${}^{2}$)

**Table 2 sensors-21-00595-t002:** Summary of this approach advantages.

Technique	Computation Time	Complexity Environment	Constraints	Result
Bézier	low-medium	low-medium	difficult	$C^{n}$ and $G^{n}$ continous
Linear MPC	medium	high	easy	fit vehicle dynamics with
				a normal performance
Non-Linear MPC	high	high	easy	fit very well
				vehicle dynamics
Proposed approach (hybrid)	medium	medium	easy-medium	average and fast performance

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
