# Peer review of "A Hybrid Planning Approach Based on MPC and Parametric Curves for Overtaking Maneuvers"

_sensors, 2021, doi:10.3390/s21020595_

Round 1
Reviewer 1 Report
The paper presents an approach on how to implement steering and speed control for automated driving in highway type conditions. Automated driving solution must be able to control the vehicle in real-time and react to all possible conditions. This paper presents evaluation of a solution which combines Bazier-curve based trajectory planning and MPC for obstacle avoidance including overtaking related lane changes. This combination can answer the aforementioned requirements.
Both of the utilised approaches are commonly used in the area and the authors have previously presented how they can be combined. This paper, to the extent that I understand, moves this area forward by presenting an evaluation where an actual vehicle run under the control of a system built on this approach (obstacles were simulated, if I read correctly). Simulated tests are reported as well.
The paper has a reasonably clear structure. However, language needs proof-reading as at times grammatical errors make reading cumbersome. Various figures included are all valuable and help communicate the content.
One bigger edit that could be valuable is explicitly stating what is new in this paper. I had to check the 2018 paper of the authors and compare it to this manuscript to see, what is new here.
I also urge the authors to make sure the text introduces all the variables and terms that are used. I'm not sure if, for example, "Lat. Error", "Long. Error" and K found in Figure 4 are explained.
I would also like to see a clear description of what information the solution takes in, i.e., how positions and velocities of other vehicles and other obstacles are given to the generator and if some other information travels from perception part forward.
Minor issues to fix include: keywords are missing, dimensions of Twizy have width and length in wrong order.
Overall, this appear to me as a small but noteworthy step forward in the area and is worth publishing after language and small presentation issues have been polished. However, adding more systematic reporting of the evaluations supporting statements made around line 343, e.g., how many different conditions was the system exposed to and how long a distance has it driven could be worth the effort.
Author Response
We would like to express our appreciation for your time spent in carefully reviewing our article and for your constructive and insightful comments. An extensive revision of the original manuscript has been performed, carefully following the suggested changes. We have also carefully proofread the revised manuscript and tried to eliminate all spelling and grammar errors. Please find below our detailed response to each one of your specific comments.
Reviewer 1 report:
The paper presents an approach on how to implement steering and speed control for automated driving in highway type conditions. Automated driving solution must be able to control the vehicle in real-time and react to all possible conditions. This paper presents evaluation of a solution which combines Bazier-curve based trajectory planning and MPC for obstacle avoidance including overtaking related lane changes. This combination can answer the aforementioned requirements.
Both of the utilised approaches are commonly used in the area and the authors have previously presented how they can be combined. This paper, to the extent that I understand, moves this area forward by presenting an evaluation where an actual vehicle run under the control of a system built on this approach (obstacles were simulated, if I read correctly). Simulated tests are reported as well.
Answer: Thank you for your comments. Some previous works from the authors have considered Bezier curves, and the first approach with MPC + parametric trajectories was published in REF 45. Only simulation results without moving obstacles were considered in this previous work. In this paper, our solution is extended to combine the passenger’s comfort associated with the smoothness of Bezier curves and MPC to react against unexpected conditions. In this work, real tests, using communication protocols, have been implemented for the overtaking maneuver with two connected and automated vehicles in complex scenarios with static and moving obstacles (at 60 kph). The simulated result in section 5.3. Overtaking scenarios using virtual environments are considered simulated overtaking but with three vehicles: the overtaken, the overtaking car, and another coming in the opposite direction, to validate our proposal in a complex scenario. Section 5.1 and 5.2 are using real vehicles. A better explanation is included in the introduction of the paper.
REF 16: Lattarulo, R.; Hess, D.; Pérez, J. A Linear Model Predictive Planning Approach for Overtaking Manoeuvres Under Possible Collision Circumstances. IEEE Intelligent vehicles symposium (IV) 2018.
The paper has a reasonably clear structure. However, language needs proof-reading as at times grammatical errors make reading cumbersome. Various figures included are all valuable and help communicate the content.
Answer: The comment has been considered with an extensive language review process and it was complemented with the comments of reviewers 2, 3, and 4.
One bigger edit that could be valuable is explicitly stating what is new in this paper. I had to check the 2018 paper of the authors and compare it to this manuscript to see, what is new here.
Answer: Thanks for your comment. A better explanation is included in the introduction of the paper.
I also urge the authors to make sure the text introduces all the variables and terms that are used. I'm not sure if, for example, "Lat. Error", "Long. Error" and K found in Figure 4 are explained.
Answer: Thanks for your comments. A better explanation of the variables used was introduced in the text. See section 3.3.
I would also like to see a clear description of what information the solution takes in, i.e., how positions and velocities of other vehicles and other obstacles are given to the generator and if some other information travels from perception part forward.
Answer: A paragraph was added in section 3.4, and this specifies the values used to model the obstacles.
Minor issues to fix include: keywords are missing, dimensions of Twizy have width and length in wrong order.
Answer: More relevant keywords were added. The changes in the table 1 have been highlighted and the order of width and length has been modified.
Overall, this appear to me as a small but noteworthy step forward in the area and is worth publishing after language and small presentation issues have been polished. However, adding more systematic reporting of the evaluations supporting statements made around line 343, e.g., how many different conditions was the system exposed to and how long a distance has it driven could be worth the effort.
Answer: The authors believe further tests are needed as well as including multiple real vehicles. The authors are working to do more tests soon (during the next months) with these conditions.

Reviewer 2 Report
Overall, the English / grammar needs review and extensive editing for readability. Also, more details on the data acquisition system on the real vehicle should be provided.
- Line 20: “more than 50 result injured” is unclear.
- Line 23: “high concerned level” is unclear.
- Lines 27-29: How do these three automation levels relate to the more recently published SAE “Levels of Driving Automation”?
- Line 38: “resume” should be “review”
- Line 59: remove apostrophe from “driver’s”
- Line 73: Sentence wording is unclear
- Line 147: Sentence wording is unclear
- Line 194: Sentence wording is unclear
- Section 3.4 – The advantage of MPC is that it is generalizable. However, these constraints seem specific to the overtaking problem. How will this algorithm work in other types of obstacle avoidance and/or lane keeping scenarios?
- Line 242: Sentence wording is unclear
- Section 4.3 should include some description of the data acquisition system on the real vehicle. How where the data used to generate Figures 8-11 acquired.
- Table 2: “hybrid” is misspelled
- Lines 373-375: Sentence wording is unclear.
Author Response
We would like to express our appreciation for your time spent in carefully reviewing our article and for your constructive and insightful comments. An extensive revision of the original manuscript has been performed, carefully following the suggested changes. We have also carefully proofread the revised manuscript and tried to eliminate all spelling and grammar errors. Please find below our detailed response to each one of your specific comments.
Reviewer 2 report:
Overall, the English / grammar needs review and extensive editing for readability. Also, more details on the data acquisition system on the real vehicle should be provided.
Answer: Thanks for your comment. An extensive language review was performed. It was complemented with the comments of reviewers 1, 2, 3, and 4.
Line 20: “more than 50 result injured” is unclear.
Answer: The comment has been considered and the sentence was complemented.
Line 23: “high concerned level” is unclear.
Answer: The comment has been considered and the sentence was structured in a different way.
Lines 27-29: How do these three automation levels relate to the more recently published SAE “Levels of Driving Automation”?
Answer: The comment has been considered and the sentence was structured in a different way to avoid misunderstandings.
Line 38: “resume” should be “review”
Answer: The error was considered and fixed.
Line 59: remove apostrophe from “driver’s”
Answer: The change was considered.
Line 73: Sentence wording is unclear
Answer: The comment has been considered and reviewed.
Line 147: Sentence wording is unclear
Answer: The sentence was modified to be clear for the reader.
Line 194: Sentence wording is unclear
Answer: The sentence was modified.
Section 3.4 – The advantage of MPC is that it is generalizable. However, these constraints seem specific to the overtaking problem. How will this algorithm work in other types of obstacle avoidance and/or lane keeping scenarios?
Answer: A paragraph was added at the end of section 3.3 explaining this statement considering the nominal planner and the MPC.
Line 242: Sentence wording is unclear
Answer: The sentence was modified
Section 4.3 should include some description of the data acquisition system on the real vehicle. How where the data used to generate Figures 8-11 acquired.
Answer: A paragraph was added in section 3.4, and this specifies the values used to model the obstacles.
Table 2: “hybrid” is misspelled
Answer: The comment was reviewed (highlighted, table 2).
Lines 373-375: Sentence wording is unclear.
Answer: The paragraph has been written again to make it simpler and clear for the readers.

Reviewer 3 Report
This study tries to proposed a hybrid planning method (comprehensively using Bezier curves and MPC) to solve the obstacle avoidance problems by overtaking. This paper is generally well-written with clear organization. But, to be honest, this idea is not novel in the literature according to my knowledge. Using ‘Bezier curve’ together with ‘MPC’ to search the related studies in Google Scholar, the results show that there are many papers comprehensively using Bezier curves and MPC to solve the obstacle avoidance problems. Therefore, I am sorry I cannot accept the stated innovation in this manuscript.
Based on the previous knowledge in the literature, the authors should compare the performance of the proposed method with the other similar approaches to convince the advantages of the proposed approach. Using just a simple table (i.e., Table 2) is far from enough because: (1) the results of the compared methods in the table are not presented for comparison, (2) comparison with the state-of-the-art approaches is missing. Without these detailed results, the conclusions presented in table 2 are not convincing.
The frequently mentioned emergency braking scenario is not examined in this study.
The first two sections of the manuscript are kind of distractive and not strictly focused on the examined topic in this study. A better organization of these two sections is suggested to avoid distractive contents.
Some minor comments:
The keywords are missing.
It is not reasonable to name it as ‘high speed’ with a maximum speed lower than 20 m/s.
The English needs to be polished. There are some typos, e.g., Line 38 a good resume of the of motion planning. And some syntax errors, e.g., this the main contribution of this work. Please carefully proofread the manuscript before re-submission to avoid these language problems.
Author Response
We would like to express our appreciation for your time spent in carefully reviewing our article and for your constructive and insightful comments. An extensive revision of the original manuscript has been performed, carefully following the suggested changes. We have also carefully proofread the revised manuscript and tried to eliminate all spelling and grammar errors. Please find below our detailed response to each one of your specific comments.
Reviewer 3 report:
This study tries to proposed a hybrid planning method (comprehensively using Bezier curves and MPC) to solve the obstacle avoidance problems by overtaking. This paper is generally well-written with clear organization. But, to be honest, this idea is not novel in the literature according to my knowledge. Using ‘Bezier curve’ together with ‘MPC’ to search the related studies in Google Scholar, the results show that there are many papers comprehensively using Bezier curves and MPC to solve the obstacle avoidance problems. Therefore, I am sorry I cannot accept the stated innovation in this manuscript.
Answer: Thanks for your comments. It is true that researching in google scholar or other sites, there many results using parametric curves solutions and MPC for obstacle avoidance. However, these approaches use both techniques separately, for path planning and control stages, respectively. Other authors use only parametric curves, o lineal and non-lineal MPC for trajectory generation. In this work, we propose a MPC method in a holonomic model (lateral and longitudinal actions are decoupled naturally in the vehicle) along a merging process with the Bézier curves. We ensure the response time of demanding maneuvers, such as obstacle avoidance, lane change, and overtaking. This approach considers the vehicle system as close as possible to the real vehicle's dynamics. The prediction horizon is used to prevent future risky situations of the automated vehicle and other participants. Our approach is a hybrid solution for the trajectory planner. From our knowledge, it is not implemented in this way in the literature. It is our novel contribution, which is tested in simulated and real connected and automated vehicles.
Based on the previous knowledge in the literature, the authors should compare the performance of the proposed method with the other similar approaches to convince the advantages of the proposed approach. Using just a simple table (i.e., Table 2) is far from enough because: (1) the results of the compared methods in the table are not presented for comparison, (2) comparison with the state-of-the-art approaches is missing. Without these detailed results, the conclusions presented in table 2 are not convincing.
Answer: In some works, we have used only Bezier curves planning. Other approaches use MPC, as we mentioned in the introduction. However, it is out of scope in our work here, to perform a cuantitative comparison with the other solutions. Because several parameters can be adapted in different ways (e.g. there are many N-Bézier Curves Considered for Comfort and Safety, we compared 3, 4 and 5 order curves for different urban scenarios in “Lattarulo, R.; González, L.; Martí, E.; Matute, J.; Marcano, M.; Pérez, J. Urban Motion Planning Framework Based on N-Bézier Curves Considering Comfort and Safety. Hindawi Journal of advanced transportation 2018, pp. 1 – 13.”). It is the reason that we added the table 2, to compare qualitatively the advantages to use this hybrid solution, proposed in this work, but without any numerical comparison, because it is not part of our aim. The results are part of the Ph.D. thesis of the main author, in the framework of EU projects, which have already been finished.
The frequently mentioned emergency braking scenario is not examined in this study.
Answer: Emergency braking is considered in our design, and some results show the reduction of speed in some conditions. However, we remove this work from the abstract.
The first two sections of the manuscript are kind of distractive and not strictly focused on the examined topic in this study. A better organization of these two sections is suggested to avoid distractive contents.
Answer: Thanks for your comment. We keep the general organization of the paper because other reviews considered well structure. We improve the content in order to focus more on the ideas presented in our work.
Some minor comments:
The keywords are missing.
Answer: More relevant keywords were added (lines 13 – 14).
It is not reasonable to name it as ‘high speed’ with a maximum speed lower than 20 m/s.
Answer: Thanks for your comment. We consider high speed because the platforms used were two automated Renault Twizys, which their max. speed (from the manufacturers) is around 80 kph (approx. 22 m/s).
The English needs to be polished. There are some typos, e.g., Line 38 a good resume of the of motion planning. And some syntax errors, e.g., this the main contribution of this work. Please carefully proofread the manuscript before re-submission to avoid these language problems.
Answer: Thanks for your comment. The comment has been considered with an extensive language review. Thanks for your time reviewing our work, the author appreciates it.

Reviewer 4 Report
My comments are in the attachment.

Author Response
We would like to express our appreciation for your time spent in carefully reviewing our article and for your constructive and insightful comments. An extensive revision of the original manuscript has been performed, carefully following the suggested changes. We have also carefully proofread the revised manuscript and tried to eliminate all spelling and grammar errors. Please find below our detailed response to each one of your specific comments.
Reviewer 4 report
General: Description of the applied algorithm is significantly limited and consequently unclear. The article include many ambiguities related to the description of the control algorithm and its implementation.
Answer: Thank you for your comments. We updated the paper, based on the comments from the reviewers. These changes have been highlighted in yellow in the manuscript.
Page 2, line 63: “space and time gaps must be temporally blocked” - it is not clear what space and time gaps and what does it mean that they are temporally blocked.
Answer: These sentences was modified, including space-time with other moving vehicles around.
Page 2, line 65: “blockage of the space” - not clear
Answer: The “space” used in this sentence is related to cooperative manuvers.
Page 3, line 89: “at low speeds” - it should be clarified what does it mean “low”
Answer: Thanks. We added “less than 15 kph”, based on [35].
Page 4, line 120: “related merging process” -> “related to the merging process”?
Answer: Thanks for your comments. We have corrected it.
Page 5, line 154: “longitudinal and lateral dynamics” - with respect to road or vehicle ? Only on the basis of the remainder of the article it can be concluded that it is probably with respect to the vehicle.
Answer: You are right. It is with respect to the vehicle.
Page 6, line 173, Equation (5): Directions of x, y, z axis were not defined.
Answer: Thanks for the comments. The a_x, a_y and a_z are the accelerations in the three axes. A sentence was included.
Page 6, line 173, Equation (5): On what basis the second equation was evaluated based on v_lon ?
Answer: The first equation of (5) has a higher influence of lateral acceleration (a_y) than the others. It is why, we consider than: a_y = v_long * k, where K is the curvature of the road.
Page 6, line 183, Equation (9): In what range the lateral road coordinated is defined from -0.5Rw or from zero. It is not clear why 3/2Rw is used ?
Answer: The 3/2 is related to the upper bound of the road. A better explanation of the variable ${d}_{lat}$ is included.
Page 7, line 190: “it is minimized the function” → “it minimizes the function” ?
Answer: Thanks for your comments. We have corrected it.
Page 7, line 191: It was not explained in the article why j_lon and a_lat are defined as control vector or control variables.
Answer: Thanks for the comment, the explanation of using both values was added in section 3.2.1 and 3.2.2.
Page 7, line 203: It was not described in details, by mathematical formulas, what is meant by “Nominal Trajectory Calculator”
Answer: Thanks for your comments the information was added and a reference was included.
Page 7, line 204: “lateral error, angular error, curvature” were not defined in the manuscript.
Answer: The angular error is the angle between the vehicle’s axis and the trajectory, measured in degrees. The lateral error is the deviation, in meters, of the front of the vehicle to the same trajectory. And the curvature is the amount by which a curve deviates, as it is defined in REF 45.
Page 7, line 205: It was not described in details, by mathematical formulas, what is meant by “MPC calculator”. Explanation: “is in charge of modeling the constraints and the reference of the MPC” is not clear.
Answer: A link to the section 3.4 was added. This section explains the calculation performed to adapt the reference and constraints.
Page 7, line 206: It was not described in details, by mathematical formulas, what is meant by “MPC output”
Answer: A link to the section 2.3.2 was added.
Page 7, line 207: “generate the states” - please clarify what states.
Answer: Further information was added in the phrase.
Page 7, line 211: “control error” were not explained, defined in the manuscript
Answer: It was a misunderstanding. The phrase was modified and fix considering previous comments.
Page 8, Figure 4: Figure 4 is not clear at all, e.g.: Perception Communication”, “Buffer”, “Lateral Control”- mathematical description of control algorithm should be given, “Longitudinal Control” lack of mathematical description.
Answer: Thanks for your comment. This architecture was modified from other previous works, to adapt the Bezier trajectories + MPC proposed in this work. The paragraphs were modified to introduce this concept, and better understand Fig. 4. Most of the concepts used were explained better, giving more details about the variables used. The references were updated, and in some cases added.
Page 10, line 248: the key elements and mathematical description of the reactive controller should be recalled.
Answer: Thanks for your comments. Details of this controller are presented in [46]. The sentence “based on lateral and angular error and the curvature of the trajectory” was added. The controller is presented in equation (14).
Page 10, line 256: control variables were not defined.
Answer: The variables: lateral and angular error and the curvature, were added in the text. See equation (14).
Page 10, line 260, Equation (12): Mathematical description and justification of H(s) should be
given.
Answer: The phrase was written again to avoid misunderstanding.
Page 10, line 260, Equation (13): Please explain why delay of H(s), and dynamics of the vehicle were not taken into account in distance Db.
Answer: The phrase was written again to avoid misunderstanding. This considers previous comment.
Page 11, line 263: lateral error, angular error and curvature were not defined.
Answer: This explanation was added in section 3.3
Page 11, line 264, Equation (14): It is not clear what “C v_lat” refers to. Please explain how it is applied in the control algorithms?
Answer: with explanation of the control variables, the application of this controller is clear. See reference [46] for more details.
Page 11, line 270: Details about applied fuzzy controller should be recalled including definition of speed error.
Answer: Thanks for the comment. It was a mistake in the reference used, we updated. A new sentence was added in the text, in order to better explain the longitudinal fuzzy controller.
Longitudinal fuzzy logic controllers can be easily and intuitively designed using human experience, as in \cite{PEREZ20131024}. A fuzzy controller was used to control the longitudinal domain. The membership functions were defined with current speed, where three different membership functions were defined for low, medium, and high, and the speed error, where the membership functions are negative, central, and positive, as in \cite{PEREZ20131024}. The output of the fuzzy controller is the normalized action on the throttle and brake pedals, defined in the range [-1,0) for the brake, and in the range (0, 1] for the throttle.
Page 11, line 11: It is not clear if this paragraph refers to simulation or experimental results.
Answer: Thanks for the comment, the parts related to simulation and experimental results were specifically added to avoid misunderstandings.
Page 12, line 295: In the reviewer’s opinion this paragraph should be moved to the section of Experimental results.
Answer: The information was moved to section 5.2.
Page 12, line 305: Value of control horizon evaluated for the low-medium speed was not given which make it difficult to analyze the results.
Answer: This information was added in the beginning of section 5.
Page 12, line 326: Value of control horizon evaluated for the medium-high speed was not given which make it difficult to analyze the results.
Answer: This information was added in the beginning of section 5. They are equal to the ones used in the low speed test case.
Page 16, line 350: “10 miliseconds” - Results of computational time should be always compared with exact specification of the computational system.
Answer: The hardware used was mentioned.
Page 16, line 350: The reviewer wonders if it would be better to call the applied integrators chains as kinematic model, not dynamics.
Answer: The review was considered, and the sentence was changed
Page 16, line 358: Method of communication with other vehicles was not described in the manuscript.
Answer: A paragraph was added in section 3.4 to explain this topic.

Round 2
Reviewer 3 Report
Although I cannot completely agree on the novelty of this paper, it is worth publishing in Sensors for readers from both research and industry communities to learn the effectiveness of Bezier curve + MPC in overtaking maneuvers. I do like the presentation of this paper, which gains a lot on my final decision. Besides, the following two recently-published related work (with one published in Sensors) should be included in the reviewed literature for a better picture: Hybrid Path Planning Combining Potential Field with Sigmoid Curve for Autonomous Driving and Adaptive Potential Field-Based Path Planning for Complex Autonomous Driving Scenarios. Please be careful on the novelties of your future submission and looking forward to your better contributions.
Author Response
Dear reviewer, we would like to thank your comment. Both references have been included to improve the background information of our work.

Reviewer 4 Report
Comments are given in the attached PDF file.

Author Response
Thanks for your comments. The answer to each comment is in the attached pdf
